# An Experimental Model of Proton-Beam-Induced Radiation Dermatitis In Vivo

**DOI:** 10.3390/ijms242216373

**Published:** 2023-11-15

**Authors:** Viktoriia A. Anikina, Svetlana S. Sorokina, Alexander E. Shemyakov, Elizaveta A. Zamyatina, Iuliia S. Taskaeva, Polina O. Teplova, Nelli R. Popova

**Affiliations:** 1Institute of Theoretical and Experimental Biophysics of the Russian Academy of Sciences, 3 Institutskaya St., Pushchino 142290, Russia; viktoriya.anikina@list.ru (V.A.A.); sorokinasvetlana.iteb@gmail.com (S.S.S.); alshemyakov@yandex.ru (A.E.S.); sonyoru162@gmail.com (E.A.Z.); 2Branch “Physical-Technical Center” of P.N. Lebedev Physical Institute of the Russian Academy of Sciences, 2 Akademichesky Proezd, Protvino 142281, Russia; 3Research Institute of Clinical and Experimental Lymphology—Branch of the Institute of Cytology and Genetics, Siberian Branch of the Russian Academy of Sciences, 2 Timakova St., Novosibirsk 630060, Russia; inabrite@yandex.ru; 4Institute of Cell Biophysics of the Russian Academy of Sciences, 3 Institutskaya St., Pushchino 142290, Russia; p.o.teplova@gmail.com

**Keywords:** ionizing radiation, noncancer radiation effects, radiation dermatitis, skin toxicity, proton therapy, animal model

## Abstract

Radiation dermatitis (RD) is one of the most common side effects of radiation therapy. However, to date, there is a lack of both specific treatments for RD and validated experimental animal models with the use of various sources of ionizing radiation (IR) applied in clinical practice. The aim of this study was to develop and validate a model of acute RD induced using proton radiation in mice. Acute RD (Grade 2–4) was obtained with doses of 30, 40, and 50 Gy, either with or without depilation. The developed model of RD was characterized by typical histological changes in the skin after irradiation. Moreover, the depilation contributed to a skin histology alteration of the irradiated mice. The assessment of animal vital signs indicated that there was no effect of proton irradiation on the well-being or general condition of the animals. This model can be used to develop effective therapeutic agents and study the pathogenesis of radiation-induced skin toxicity, including that caused by proton irradiation.

## 1. Introduction

Cancer treatment is currently mostly based on surgery, chemotherapy, and radiation therapy [1,2,3]. For more than 60% of the patients with malignant tumors, radiotherapy is an integral component of treatment, either in combination with surgery or chemotherapy or on its own [4]. In particular, proton therapy has significantly developed clinically in the last two decades, although the method itself was discovered more than half a century ago [5,6,7]. Proton therapy has been found to reduce the likelihood of early or late radiation complications in healthy tissues, including secondary radiation-induced tumors [8]. The importance of this issue cannot be overstated in terms of radiotherapy for children and young patients. Despite technological advances, proton therapy, as well as conventional gamma therapy, has a common side effect known as radiation dermatitis (RD) [9,10,11].

During the course of conventional radiotherapy, 95% of patients develop some grade of RD, which is a toxic skin reaction to ionizing radiation (IR) [10,12]. A significant portion of the radiation dose in radiotherapy is absorbed by the skin, especially when using conventional X-ray or gamma-radiation therapy. Measures that reduce the dose absorbed by the skin can include the use of boluses and different methods of dose distribution depending on the type of radiation. Hadron radiotherapy achieves efficient targeting of tumor tissue with low impact on healthy peritumoral tissues [13,14].

Today, there are no specific treatments for RD. Therefore, a crucial task for modern biomedicine is the development of agents to prevent and treat radiation burns [15,16]. To assess the effectiveness of new medications, we need a reproducible model of RD induced using the different types of IR (X-rays, gamma rays, protons, and accelerated ions) applied in radiation therapy and having differences in relative biological effectiveness in relation to the skin. There are number of studies published regarding the radiation dermatitis model in mice induced by single or fractionated X-ray irradiation at doses ranging from 20 to 60 Gy [17,18,19,20]. Most commonly, the target area for simulating RD in such experiments is the leg, back, or flank of the animal, although there are studies on radiation-induced damage to the feet [21]. It is also worth noting the studies on the impact of β-radiation and comparing the effects of γ and β radiation on the skin of miniature pigs [22,23]. Furthermore, this year saw the publication of unique studies describing dosimetric characteristics and the analysis of acute and late effects of normal tissue damage in response to single and fractionated proton irradiation [24,25]. However, the research is currently limited on the effects of proton irradiation on the skin. 

The aim of this study was to develop an RD model on the dorsal side of the mouse’s body after a single local exposure to proton radiation using focused delivery at the Bragg peak with an energy of 88 MeV. 

## 2. Results

A series of experiments were performed to select the optimal conditions for a mouse model of proton-radiation-induced RD. When developing the experimental model of RD, the following parameters of the beam and experimental conditions were taken into consideration: absorbed dose, position of the energy release peak of the monoenergetic proton beam, positioning and fixation of animals, method of depilation or its absence, duration of latent period and stages of RD, model reproducibility (% of animals with expected parameters of RD), assessment of vital signs (weight change, hematological blood test, pathomorphological examination of internal organs (thymus, spleen)), and animal survival. Experimental groups of animals were divided into subgroups: nondepilated (CND—control nondepilated animals, ND30—nondepilated animals that were irradiated with a dose of 30 Gy, ND40—nondepilated animals that were irradiated with a dose of 40 Gy); depilated via a physical method (trimmer) (CD—control depilated animals via physical method, D30—depilated animals that were irradiated with a dose of 30 Gy, D40—depilated animals that were irradiated with a dose of 40 Gy), and depilated via a chemical method (cream) (CChD—control animals depilated via chemical method, ChD50—depilated animals that were irradiated with a dose of 50 Gy). As a result of the experiments, a model of proton-induced RD in mice was developed that meets the following requirements:Location of the energy release peak of the proton beam is directly in the skin of the animal in order to reduce radiation damage to the underlying tissues;Model reproducibility of more than 90%;Absence of a long latent period (up to 2 weeks);Pronounced staging of RD formation;100% survival of animals within 30 days of exposure.

We think that the developed model of radiation-induced skin damage will help to expand the knowledge about the pathogenesis of proton-radiation-induced RD and can be used to evaluate the effectiveness of preventive and therapeutic agents.

### 2.1. Effect of the Depilation Method on the Skin of Mice

When modeling skin wounds in animals, the hair cover is usually removed. To date, there are no detailed unified criteria for choosing the method of animal depilation, particularly in RD modeling. In this regard, we investigated the effects of physical (trimmer) (CD) and chemical (depilatory cream) (CChD) hair removal methods on the skin of animals that had not been exposed to proton radiation (Figure 1). The skin reaction and hair regeneration after different depilation methods were recorded for 10 weeks using a photo camera and visual assessment. 

As can be seen in Figure 1A, the animals subjected to physical depilation (CD) did not show any skin reactions one day after hair removal. In the group of animals subjected to chemical depilation (CChD), patches, redness, and abrasions were observed on the skin (Figure 1B). By the third week after depilation, hair regrowth and no visible skin reactions were observed in both groups of animals. It should be noted that the rate of hair regrowth after chemical depilation was slightly higher than that after physical depilation. However, starting from the fourth week, there were no visible difference in skin reactions and hair regrowth between CD and CChD. Thus, in animals after chemical and physical depilation, complete hair regeneration was observed by the fourth week. However, one day after chemical depilation, the animal skin had wounds, which, in turn, can increase the severity of skin toxicity and healing time in RD modeling. 

Thus, we think that the physical method of depilation should be preferred in RD modeling, taking into account the absence of initial skin reactions in animals, potential contribution of depilation to the severity of the wound, and the rate of skin recovery in irradiated mice. The effect of depilation method on the severity of the RD grade and healing time of radiation-induced skin damage in experimental animals are considered in Section 2.4.

### 2.2. Assessment of the Physical Characteristics of a Proton Beam for Producing a Mouse Model of Radiation Dermatitis

At this stage of the work, the optimal proton beam energy was selected to create a mouse model of RD. The minimum energy of the proton beam released from the accelerator in the Prometheus proton therapy complex is 30 MeV. At this energy, the depth of particle penetration into soft tissues is less than 10 mm, while the width of the energy release peak is less than 0.5 mm, which causes significant difficulties both in dose estimation and irradiation. At this energy, a 0.2 mm error in animal positioning or beam approach can result in a dose delivery error of 50–80%, which is unacceptable for this type of experiment. To avoid these inaccuracies, a Plexiglas phantom filled with water was placed in the beam path. The phantom was 60 mm thick, which, accounting for the Plexiglas, corresponded to a water equivalent thickness (WET) of 61.8 mm. The position of the peak was checked using a parallel plate chamber, taking into account the thickness of the outer wall of the chamber placed behind the water phantom, where the animals were subsequently fixed. The peak position was checked in the energy range of 87 MeV to 92 MeV in 0.1 MeV increments. Calculations performed in the planning program (Protom Planner ver. 2.14.32, “Protom” JSC, Protvino, Russia) that is a part of the Prometheus complex showed that the energy required to pass this phantom and stop the maximum energy release on the skin of the animal was about 88 MeV (Figure 2).

Consequently, at energy of 88 MeV, the peak width at 90% of the dose level was 1.2 mm, which allowed us to plan and perform irradiation with a skin dose accuracy of at least 10%. The value of linear energy transfer (LET) for proton radiation was approximately 5–10 keV/μm. Since the transverse size of the beam was about 8 mm when it was released from the accelerator and an irradiation field of 10 × 10 mm was required when working with animals, plans for uniform irradiation via proton beam scanning over a given area were created in Prometheus complex software ("Protom Planner" ver. 2.14.32, “Protom” JSC, Protvino, Russia). 

### 2.3. Effect of Animal Positioning on the Absorbed Proton Dose Distribution

Taking into account the selected optimal values of energy and irradiation field, a dosimetric study involving a biological object (mouse) was carried out. The experimental animal was euthanized via cervical spine dislocation. After that, a skin flap was cut off from the animal’s back, and an EBT3 dosimetry film was placed over and under it (Figure 3). This procedure is necessary for the experimental control of proton beam energy since it is difficult to account for the real thickness of the epidermis and hair of the animal (if there is no preliminary depilation) when calculating. The animal was fixed in an upright position on a medical thermoplastic platform using soft clamps. The objectives of this stage of the work were to assess the effect of positioning the animals at 90° and 45° angles relative to the proton beam hitting the animal skin and to check the position of the energy release peak on the animal skin at an energy of 88 MeV.

At energy values of 88 MeV, the maximum energy release was located directly in the epidermis and dermis and did not affect underlying tissues and organs. The dosimetric assessment using the EBT film placed under the skin of the animal showed no significant difference in the dose distribution at two variations in the animal position: 45° and 90°. However, from the technical point of view, the optimal position is the 90° angle, which was used in the RD modeling in this work.

### 2.4. Clinical Changes in the Skin of Mice after Local Proton Radiation

At this stage of the work, the severity of the RD grade and the time of spontaneous wound healing of radiation-induced skin damage, depending on the absorbed dose and presence/absence of the hair coat as well as the method of depilation, were studied. The groups of animals were irradiated at doses of 30 Gy (ND30, D30), 40 Gy (ND40, D40), or 50 Gy (ChD50). Figure 4 schematically represents the time dependencies of RD grade formation according to the RTOG classification in the experimental groups of animals at 7, 14, and 21 days after exposure to proton radiation at a dose of 30 Gy, 40 Gy, or 50 Gy.

It was shown that on the seventh day after a single exposure to local proton radiation at doses of 30 and 40 Gy, all animals that were not depilated (ND30, ND40) had no skin reaction (Grade 0) to proton radiation. In the D30 experimental group, 63% of mice showed no skin reactions, representing Grade 0, and 37% were Grade 1. In D40, 44% of mice showed no skin reactions to proton irradiation (Grade 0). The other animals in this group had RD Grades 1 to 3 with Grade 1 being predominant, which was noted in 31% of the animals. In the ChD50experimental group, 55% of mice showed no skin reactions to proton irradiation, and 45% of the animals had skin toxicity, represented by Grade 1. By the 14th day after exposure to proton radiation, it was found that 12.5% and 27% of the mice in ND30 and ND40, respectively, had no skin reactions. The other animals in these groups showed RD Grades 1 to 3 with a prevalence of Grade 1. In experimental groups D30 and D40, more than 50% of the mice were Grade 2, and Grade 3 was observed in 9% and 15% of the animals. The most severe skin reactions were seen in ChD50: 67% of the mice showed Grade 4 skin reactions accompanied by ulceration and necrosis, and 33% of the mice showed radiation-induced skin toxicity corresponding to Grade 3. It was found that by the 21st day after proton exposure, the animals in all experimental groups had radiation-induced skin toxicity corresponding to Grades 2–4 according to the RTOG classification. Only in experimental group ND30 did a larger proportion of animals (60%) have RD Grade 2, while in the other groups, the animals showed more severe skin wounds corresponding to Grade 3 and Grade 4. Only in D30 did 100% of the animals show radiation-induced skin toxicity corresponding to Grade 3. In experimental groups ND40 and D40, regardless of whether the hair was removed, the majority of animals (80% in ND40; 75% in D40) were Grade 3, while the other animals in these groups were Grade 4 of RD. In the experimental group ChD50, by the 21st day, the number of animals categorized as Grade 4 RD increased to 89%.

Starting from day 21, the assessment of the animals’ skin condition was performed once a week. At this stage of RD development, visual assessment of the severity degree is difficult due to the ongoing reparative process. The duration of observation was 70 days because this time is enough to monitor the course of acute RD. At this stage of the work, we carried out visual assessment of the development of the reparative stage of RD in mice depending on the absorbed dose and method of depilation.

#### 2.4.1. Study of the Development of Radiation-Induced Skin Damage in Nondepilated Mice

Figure 5 shows the dynamics of spontaneous wound healing of radiation-induced skin damage in nondepilated mice (ND30, ND40) before irradiation and 21 and 70 days after irradiation compared to the control (CND).

As shown in Figure 5, by day 21 following a single local proton irradiation on the dorsal side of the body, the area of damage increased threefold with the increase in dose from 30 Gy to 40 Gy. By day 70, ND30 mice showed completed epithelialization and hair regrowth processes, while ND40 mice showed nonhealing skin lesions that presumably proceeded to chronic RD. It is known that the acute phase of RD transitions to chronic after 90 days.

#### 2.4.2. Study of the Development of Radiation-Induced Skin Damage in Physically Depilated Mice

Figure 6 shows the dynamics of spontaneous wound healing of radiation-induced skin damage in mice depilated via a physical method (D30, D40) before proton irradiation and 21 and 70 days after irradiation compared to the control (CD).

As can be seen in Figure 6, no skin reactions were observed after hair removal using a physical method. By day 21, radiation-induced skin lesions of the same area but of different severity had formed in both groups. As in animals with preserved hair coat, the dose–response relationship was observed: with an increase in the absorbed dose, the number of animals with more severe grades of RD increased. By day 70, there was no complete healing of skin wounds and hair regrowth in experimental groups D30 and D40.

#### 2.4.3. Study of the Development of Radiation-Induced Skin Damage in Chemically Depilated Mice

Figure 7 shows the dynamics of spontaneous wound healing of radiation-induced skin damage in chemically depilated mice (ChD50) before proton irradiation and 21 and 70 days after exposure compared to the control (CChD).

Figure 7 shows that the use of a depilatory cream left cuts and erythematous spots on the skin of the animals, which further contributed to skin damage even before exposure to proton radiation. By day 21, more than 85% of the animals after a single exposure to local proton radiation at a dose of 50 Gy showed severe skin toxicity corresponding to Grade 4. By day 70, the damaged area did not reduce, and there was no hair regrowth, which resulted in the subsequent formation of chronic RD.

The duration of observation of the reparative phase after different methods of depilation or its absence was 70 days. During this period, complete healing of radiation-induced skin damage and hair regrowth were achieved only in ND30 mice, and in other groups, reparative processes continued (D30, ND40, D40, and ChD50). It should be noted that in the reparative phase as well as in the destructive one, there was a pattern: with an increase in the absorbed dose the grade of RD and the time of spontaneous wound healing of the wound increased as well.

### 2.5. Influence of Proton Irradiation and Depilating Status on Mouse Skin Histology

Skin histology was assessed on the 14th day after proton exposure in the nondepilated irradiated groups (30 and 40 Gy; ND30 and ND40, respectively), in the depilated using a physical method (trimmer) irradiated groups (30 and 40 Gy; D30 and D40, respectively), and in control nonirradiated groups (nondepilated and depilated using a physical method (trimmer, CND and CD, respectively). Figure 8 shows the representative images and results of mouse skin slide examination. 

Overall, the proton beam irradiation affected skin histology in a dose-dependent manner, with physical depilation producing additional substantial impairment. The epidermal and dermal thicknesses were significantly larger in depilated and irradiated mice (D30 and D40 groups). Moreover, the skin of the depilated and irradiated mice was characterized by a loss of follicular units, as well as by the presence of epithelial hyperplasia and hyperkeratosis. The epidermal thickness was significantly thicker in depilated and irradiated mice (4.2- and 9.4-fold increases in D30 and D40 groups, respectively, vs. CND group and 3.1- and 6.8-fold increases in D30 and D40 groups, respectively, vs. CD group). Similarly, in the depilated and irradiated groups, the dermal thickness was significantly thicker compared to that in both control groups. The number of follicular units between the CND group and irradiated groups showed approximately two- to three-fold differences. All mice irradiated with either 30 Gy or 40 Gy showed no sign of changes in the number of follicular units compared to the depilated control mice (CD group). The scores of epithelial hyperplasia and parakeratotic hyperkeratosis showed significant differences between of the control (CND and CD) and irradiated groups. 

### 2.6. Assessment of Animal Vital Signs after a Single Exposure to Local Proton Radiation

We assessed the vital signs of the experimental animals relative to intact animals according to the following criteria: 30-day survival, body weight dynamics, morphological composition of peripheral blood, and weight coefficients of internal organs (thymus, spleen).

#### 2.6.1. Study of 30-Day Survival and Body Weight Dynamics of Mice in Modeling Radiation Dermatitis

We conducted a study to investigate the effects of single local proton irradiation at doses ranging from 30 to 50 Gy on the 30-day survival (LD_50/30_) of mice, as well as changes in body weight and time of survival within 70 days after irradiation. In our work, by the 30th day, the survival in all groups of animals was 100%; moreover, all animals survived up to the end of the experiment (70 days). 

Figure 9 demonstrates the changes in body weight of the experimental animals 7, 21, and 70 days after single doses of proton radiation from 30 to 50 Gy compared to the initial body weight before exposure (day 0).

The study of body weight change showed no significant differences between the groups of nondepilated (ND30 and ND40) and chemically depilated (ChD50) animals compared to the control group (CND and CChD) (Figure 9A,C). In the D30 and D40experimental groups, a decrease in body weight of animals was observed at 7 and 21 days after a single local proton radiation exposure compared to the control group (CD), which may be associated with the peak of the destructive phase of RD development (Figure 9B). By the end of the experiment, an increase in animal body weight was observed in all groups, on average by 10%, compared to the initial body weight at the start of the experiment (day 0). 

#### 2.6.2. Effect of Proton Radiation on the Subset Composition in the Blood of Animals in Modeling Radiation-Induced Skin Damage

Hematological data (number of white blood cells, lymphocytes, granulocytes, and platelets) of the peripheral blood of the white male SHK mice before irradiation and 7, 21, and 70 days after a single exposure to local proton radiation at doses from 30 to 50 Gy compared to the control (initial value for the four classes of blood formed elements) were studied (Appendix A). Despite the significant intragroup variability, likely associated with the use of outbred nonlinear SHK mice as study subjects, the presented results reflect a number of changes. In the CChD group, increases of 41% and 37% in WBC numbers were observed at 21 and 70 days after irradiation, respectively, as well as by 86% in the ChD50 group at day 21; however, these differences were not statistically significant. The analysis of the number of lymphocytes revealed that the ChD50 group showed a 40% increase by day 7, while in the D30 group, the cell count decreased by 51%. By day 21, the number of lymphocytes increased by 33%, 36%, and 43% in the D30, CChD, and ChD50 groups, respectively. Changes in the number of granulocytes were observed only in the ChD50 group 21 days after exposure. As for statistically significant results, they were obtained only for platelets in the overall blood analysis. Thus, in the CChD group, the cell numbers increased by 144% and 120% at 21 and 70 days, respectively, while in the ChD50 group, the number of platelets increased by 160% at day 7, with the trend continuing until the end of the observation period (70 days). 

#### 2.6.3. Study of Changes in the Relative Mass of Lymphoid Organs of Experimental Animals in Modeling Radiation-Induced Skin Damage

In the study, we studied the changes in relative masses of spleen and thymus in % of experimental animal weights 7, 21, and 70 days after a single exposure to local proton radiation at doses from 30 to 50 Gy (ND30, ND40, D30, D40 and ChD50) compared to those of intact animals (CND, CD and CCh). The data are graphically presented in Figure 10 and Figure 11.

There were no significant changes in the relative mass of the spleen in either experimental group or intact mice throughout the experiment (70 days). No significant changes were observed in the thymus mass in any experimental group at 7 days after irradiation, while a decrease in organ mass was observed in all groups except ND30 and CD at 21 days compared to the control. In the ChD groups, this decrease remained significant up to 70 days. Thus, the reduction in the relative mass of the thymus was presumably associated with an increase in destructive processes and a decrease in the lymphoid cell population in response to proton irradiation, which was sustained until the end of the experiment in animals subjected to chemical depilation.

## 3. Discussion

In this study, a protocol was developed and an experiment was conducted to model proton-induced radiation dermatitis on the dorsal side of an animal for the first time. The model of local proton-induced dermatitis on the back of a mouse was obtained through a single exposure of anesthetized animals fixed perpendicular (90°) to a proton beam with an energy of 88 MeV. The absorbed dose ranged from 30 to 50 Gy, and the linear energy transfer (LET) value was approximately 5–10 keV/μm. The recommended irradiation field size is 10 × 10 mm. The developed radiation dermatitis model has several advantages for its in vivo application. Firstly, the single exposure enables the avoidance of subjecting the animal to multiple anesthesia procedures, thus minimizing the stress induced by transportation and repeated immobilization, thereby adhering to the requirements set forth in Directive 2010/63/EU of the European Parliament and Council of the European Union on the protection of animals used for scientific purposes. Secondly, the local administration directly onto the animal’s skin, facilitated by the presence of the Bragg peak, prevents damage to the spinal column and adjacent neural and muscular tissue. Thirdly, localizing the target area of radiation exposure at the center of the animal’s back allows the minimization of further animal interference in the process of applying the therapeutic agent and the wound healing stage.

When studying in vivo model wounds, it is important to minimize the contribution of depilation to skin damage. As previously noted, there are no standardized methods for animal hair depilation when modeling skin damage, specifically RD. Furthermore, there is a lack of data on the necessity of removing or preserving the hair coat of animals before modeling RD.

There are limited data on the suitability of using depilatory products on animals, but some studies have shown that both physical and chemical methods of depilation do not lead to an increase in bacterial load and prolonged wound healing in rat and mouse models of skin damage [26,27]. In a recent study [28], four methods of depilation were compared in a wound model in mice. The authors demonstrated that the sodium sulfide contained in depilatory creams has a significant negative effect on wound healing through the enhancement in local inflammatory reactions, a sharp decrease in the number of hair follicles, and the inhibition of collagen fiber regeneration. In another study [29], the authors suggested that the ingredients in depilatory creams contribute to an increase in the number of dermal CD45+ cells and mast cells in mice and influence changes in skin homeostasis. In addition, chemical depilatory agents can cause chemical burns during prolonged contact with the skin or induce individual intolerance to the components included in the composition. In a study [30], it was shown that a depilatory cream significantly induced the expression of inflammatory factors, although their levels were lower than those caused by a model wound, suggesting that this type of depilation only causes a mild form of skin damage.

Physical methods of hair removal also have their drawbacks, as any hair removal method is associated with potential operator error, leading to injuries, general discomfort, stress, and poor well-being of the animals. However, a study [31] demonstrated that depilation using physical methods did not lead to visible changes in the skin of experimental animals, while chemical methods resulted in abrasions, cuts, and redness.

In this study, we conducted an investigation into the effects of physical and chemical hair depilation methods on the skin of intact and experimental animals exposed to proton radiation. Based on the obtained results, we suggest that the physical method of depilation is preferable compared with the chemical method, as it minimizes skin damage prior to irradiation. We concluded that since the choice of depilation method in RD modeling should aim to minimize the contribution of depilation to the formation of radiation-induced skin damage, it is advisable to avoid hair removal altogether. This is supported by the observation that nondepilated animals exhibited alopecia on day 14, which may serve as an additional criterion for the development of RD.

The impact of local proton irradiation on animal skin has been examined in a limited number of studies, which makes this issue particularly relevant to modern radiation biomedicine. Additionally, no studies have been found on modeling the RD caused by proton irradiation on the back. However, in recent decades, the effects of other types of ionizing radiation on the skin have been fairly well described. For example, it was demonstrated that when BALB/c mice were irradiated with a dose of 36 Gy for three consecutive days (12 Gy/day) using a source of X-ray radiation, the peak of skin damage was observed 7–10 days after irradiation [19]. In another study [20], using the same mice strain and radiation scheme, the duration of the latent period was 7 days, and the peak development of RD occurred on the 15th day. In a study [32], when C57BL/6 mice were irradiated with high-energy electrons at doses of 20 Gy or higher, RD developed 8–12 days after irradiation and progressed to severe grades (3–4) within 3 weeks. In contrast, mice receiving a dose of 15 Gy developed only mild RD. Local irradiation of the flanks of C57BL/6 mice with X-rays at a dose of 45 Gy resulted in severe skin damage, characterized by scab formation and moist desquamation [17]. Subsequently, the authors reduced the irradiation dose to 30 Gy. It is worth noting that the assessment of skin damage in the cited studies was carried out according to international clinical scales. In this study, the RTOG scale was used for the visual assessment of radiation-induced skin damage, in which reactions of different intensities are graded from one to four [33]. In Grade 1, moderate-intensity erythema, alopecia, and dry desquamation are observed. In Grade 2, there is usually pale or bright erythema, which is usually accompanied by moderate edema. In Grade 3, erythema is accompanied by edema and moist scaling. Grade 4 is characterized by ulceration, bleeding, and necrosis.

As a result of the series of experiments, it was demonstrated that a single dose of proton beam irradiation from 30 Gy to 50 Gy led to radiation-induced skin toxicity in 100% of experimental animals. The duration of the latent period varied from 7 to 11 days depending on the absorbed dose. The duration of the destructive stage in all groups was 21 ± 3 days. In the course of development of radiation-induced skin damage, pronounced staging of RD formation (latent period, erythema, dry desquamation, wet desquamation, crusting, and epithelization) was observed in the mice. The severity of the RD grade differed among the groups, and the homogeneity of RD development in each particular group ranged from 60% to 100%. The highest grade of RD in the most animals (Grade 4 in 89% of the animals) was noted in group ChD50, in which the animals were chemically depilated and irradiated at a dose of 50 Gy. In group ND30, radiation-induced skin damage was, on the contrary, the least and corresponded to Grade 2 in 60% of the animals. It should be noted that as the absorbed dose increased, the severity of RD increased from Grade 2 to 4. In addition, the presence of a hair coat in the animals contributed to a reduction in RD grade severity, so the development of alopecia may serve as another criterion for RD evaluation. In turn, depilation increased the severity of RD. Thus, our obtained results are consistent with the literature data [17,19,20,32,33], where local exposure to ionizing radiation (IR) in doses ranging from 20 to 45 Gy resulted in a latent period duration of 7–14 days from the moment of irradiation, and the severity of RD correlated with the absorbed dose regardless of the source of IR.

Skin changes in response to IR have been well studied via histological examination. Several studies have shown that exposure to IR leads to increases in the thicknesses of the epidermis and dermis; decreases in the number of blood vessels, sebaceous glands, and hair follicle density; as well as epidermal hyperplasia and parakeratosis [34,35,36]. In our developed model of RD induced using a single local proton irradiation, an increase in absorbed dose was found to be associated with increases in the thicknesses of the epidermis and dermis, a decrease in the number of hair follicles, as well as higher degrees of epidermal hyperplasia and parakeratosis compared to those in the control group. Furthermore, depilation led to significant changes in the skin after exposure to proton radiation according to histological analysis.

To obtain reliable experimental results in animal studies, it is important to consider the animal welfare indicators that can influence the experiment outcomes [37]. One of the main physiological indicators of animal well-being is body weight. A decrease in animal body weight can occur due to stress induced by transportation, injection, and anesthesia, as well as manipulations and radiation consequences, including temporary immobilization during the session and the presence of radiation-induced damage, ultimately leading to a decrease in food intake. Studies [38,39] have noted that changes in body weight depend on several variables, such as radiation dose, experimental animal strain, and time of assessment after exposure. In [40,41], the authors did not observe any effect of total densely ionizing radiation of ^56^Fe on the body weight of C57BL/6 mice at 3, 40, and 112 days after exposure. However, in other studies [42,43], a decrease in body weight relative to the control group was observed in a rat model for up to 9 months after exposure to ^56^Fe at doses of 1–4 Gy. Additionally, when rats were irradiated with protons at doses of 7.2, 8.5, 9.2, and 11.2 Gy and X-ray at doses of 5.6, 6.3, 7.3, and 8.6 Gy to the abdominal area, there was no observed change in the body weight of the animals [44]. In our study, in experimental groups D30 and D40, a decrease in the body weight of the animals was observed at 7 and 21 days after a single local proton radiation exposure compared to the control group. By the end of the experiment, an increase in animal body weight was observed in all groups, on average by 10%, compared to the initial body weight at the start of the experiment (day 0). The tendency of increasing body weight in animals may indicate their well-being following the performed manipulations, including anesthesia, transportation, and irradiation.

The study of the effects of radiation on animal survival after exposure for 30 days (LD_50/30_) is the most widely used and well-characterized criterion for assessing acute and chronic radiation damage [45]. However, most existing studies describe the effects of total body irradiation on animal survival. Depending on the absorbed dose, tissue or organ damage can lead to acute radiation syndrome and ultimately result in death within 30 days [46]. For example, it was demonstrated that the LD50/30 for ICR mice (outbred, like SHK) after X-ray irradiation was 7.55 Gy [47]. In the case of the development of acute gastrointestinal syndrome (total dose of γ- and X-ray radiation 10–12 Gy), ICR and C57BL/6N mice die within 3.5–4 days of irradiation. It is known that body weight loss as a result of radiation exposure and the time required for weight recovery correlates with the dose. It is also known that the total proton irradiation at a dose of 8 Gy leads to 87% mortality after 30 days in ICR mice [48]. In a study [49], the 30-day survival rate after total exposure to 5.9 Gy of 1 GeV/n protons was 60%, 6.8 Gy resulted in a survival rate of 13.3%, and a dose of 7.2 Gy led to LD100/30 (death of all irradiated animals within 30 days). In this study, we demonstrated that local proton irradiation at doses ranging from 30 to 50 Gy had no impact on animal survival. Animal survival was 100% on day 30 and day 70 in all experimental groups, indicating the good state of health of the mice. These results suggest that the characteristics of radiation exposure conditions were chosen in such a way that all the dose was absorbed by the skin, while vital organs and systems were not affected.

It is well known that IR damages hematopoietic stem cells in the bone marrow, alters the ability of stromal elements in the bone marrow to support blood cell formation, and directly leads to the death of formed elements in the bloodstream [50,51]. IR causes a dose-dependent reduction in the number of circulating hematopoietic cells due not only to decreased bone marrow production but also to redistribution and apoptosis of mature hematopoietic elements. A qualitative indicator of the effect of ionizing radiation on the body may be decreases in the numbers of lymphocytes, granulocytes, and erythrocytes [41]. It is known that platelets are primarily involved in hemostasis and perform numerous other functions, including participation in inflammatory processes through complement activation, interaction with leukocytes and monocytes, involvement in the body’s defense against infection, and regulation of vascular tone [52]. In studies investigating the effects of proton irradiation on blood composition, irradiation doses of 0.5–2 Gy have been most commonly used, which are associated with fractionated therapeutic treatment of tumors, as well as cumulative doses of space radiation during prolonged missions. Thus, studies [53,54] have demonstrated dose-dependent decreases in the numbers of both leukocytes and lymphocytes in C57BL/6J mice within 24 h and on the 4th and 30th days after exposure, while the number of platelets and erythrocytes remained unchanged. In a study [54], the total body irradiation of male C57BL/6J mice with protons (150 MeV) at a dose of 0.5 Gy on both the 4th and 30th day led to decreases in the numbers of leukocytes and lymphocytes. Regarding the number of platelets and erythrocytes, no significant differences were found between the groups during either time period. In another study [55], female C57Bl/6 mice were also irradiated with cumulative protons (208 MeV) at a dose of 8.5 Gy, and on the second day after irradiation, a significant decrease in the number of leukocytes was demonstrated, followed by a decrease in the number of platelets 7–10 days later and then a decrease in the number of red blood cells approximately 2 weeks after irradiation. The recovery of all blood components began 3 weeks after irradiation. In contrast, a study [56] showed a significant decrease in the number of all types of blood cells in ICR mice, with the authors noting that the recovery of all blood components was observed after 21 days. Interestingly, in the case of ^56^Fe irradiation of mice, the number of leukocytes did not change, while the numbers of erythrocytes, hemoglobin, and hematocrit only increased on the 40th day after irradiation. We did not find any literature data on the local impact of protons on the number of peripheral blood cells in the long term after exposure. All the above-mentioned studies investigating the effects of total photon and proton radiation on the blood cell count in mice are highly contradictory, likely due to differences in the type and dose of radiation sources, mouse strains, and observation periods. 

In our work, we observed a significant intragroup variability in all studied parameters of blood cell constituents, with reliable results only reflecting changes in platelets: an increase in their quantity in the ChD50 and CChD mouse groups at 7 and 21 days, respectively. Thus, in this study, we did not observe the development of leuko-, lympho-, and granulocytopenia, which indirectly indicates the absence of radiation-induced damage to the bone marrow. We hypothesize that the method of chemical depilation contributed to the changes in the quantity of blood cells, particularly platelets, as it is known that preoperative hair removal using depilatory cream may lead to a higher risk of infection on the first day after surgery [57]. Additionally, chronic inflammation characterized by abundant neutrophil infiltration may develop in the skin [58], which, in turn, can lead to delayed wound healing and increased susceptibility to infections [59]. 

It is known that the spleen and thymus possess high radiosensitivity and are among the first to respond to the effects of IR [60]. Changes in the mass and morphology of these organs may serve as additional markers for assessing the impact of proton irradiation on the hematopoietic and immune systems. Currently, there is a lack of research on lymphoid organs, and existing literature focuses on the development of immunosuppression in various compartments of the immune system during high-dose total X-ray irradiation in the short-term period after irradiation [61,62,63]. These changes are often associated with a sharp increase in cellular destruction, leading to loss of lymphocytes, and complete inhibition of cellular mitotic activity and differentiation [64]. It was shown that total exposure to IR results in an acute or chronic inflammatory reaction, which morphologically manifests as thymus involution with progressive decrease in its mass and changes in functional activity [65]. 

This study examined the changes in the relative masses of the thymus and spleen in mice 7, 21, and 70 days after a single local skin irradiation with doses ranging from 30 to 50 Gy. The results obtained demonstrate the absence of significant changes in the relative mass of the spleen both within the experimental groups and compared to intact mice throughout the experiment (70 days). The analysis of the relative mass of the thymus revealed a significant decrease in organ mass at 21 days in all groups, except for ND30 and CD. Moreover, in the groups subjected to chemical depilation, this decrease remained significant until 70 days. We propose that the decrease in relative thymus mass is associated with enhanced destructive processes and a reduction in cellular lymphoid population in response to proton irradiation, which was maintained in animals subjected to chemical depilation until the end of the experiment. From our study, it is evident that the thymus, unlike the spleen, responds to the effects of localized proton radiation; thus, studying the changes in relative thymus mass can serve as a valuable diagnostic marker.

## 4. Materials and Methods

### 4.1. Animals

Mice were the object of the study, since it has been shown that skin toxicity in humans and mice is similar in nature and appearance [45]. White outbred male SHK mice (160 animals, 8–9 weeks old, 30–35 g) were used in the experiments. The animals were kept in polycarbonate cages with 5 individuals each with sawdust bedding in the vivarium of the ITEB RAS (Pushchino) at 22 ± 2 °C. Lighting regime of 12 h/12 h was used. The animals had free access to water and complete extruded feed for laboratory animals (OOO Provimi, Moscow, Russia). Immediately following the acclimatization session, the mice were divided into different groups.

The experiments were conducted in compliance with the ethical standards for work with laboratory animals according to the protocol approved by the ITEB RAS Commission on Bioethics and Biological Safety (No. 30/2022, 5 March 2022). All research involving animals in the ITEB RAS was carried out according to Directive 2010/63/EU of the European Parliament and Council of the European Union on the protection of animals used for scientific purposes, according to which the presence of painful procedures, anesthesia methods, the maximum expected severity of procedures, and the fate of the animals after the experiment were determined.

### 4.2. Hair Depilation in Mice

Hair depilation was performed on the dorsal side of the animals one day before irradiation using one of two methods: chemical depilation using a depilatory cream (Belle Jardin, Baniocha, Poland) or physical depilation using a pet hair trimmer (CEIEC, Jiangsu, China). The surface area was 15 × 15 mm. 

In the groups of animals subjected to chemical depilation, the hair on the dorsal side was evenly coated with depilatory cream. The duration of exposure to the cream was about 180 s. At the end of the exposure time, the hair was removed with a spatula together with the cream. After that, the skin was washed with warm distilled water (37 °C). In the group of animals that had their hair removed via shearing, the hair was shaved with a pet trimmer the day before irradiation without additional manipulation. The day after hair removal, the skin condition was captured using a camera (Nikon Corporation, Tokyo, Japan).

### 4.3. Animal Anesthesia

The mice were anesthetized with a combination of Zoletil 100 (Virbac, Carros, France) and Xyla (Interchemie, Venray, The Netherlands), which does not affect animal vital signs and is relatively safe for minor manipulations with animal biomodels [66,67,68]. Intraperitoneal injection of Xyla and Zoletil 100 at a ratio of 1:3 (concentration of 40 mg/kg) resulted in optimal anesthesia conditions, particularly, a rapid effect, the absence of movements during the procedure, and the immobilization for a minimum amount of time (20–40 min) with the preservation of vital signs.

### 4.4. Experimental Setup

Animals were irradiated at the “Physical-technical center” branch of the P.N. Lebedev Physical Institute of the Russian Academy of Science (PTC LPI RAS) (Protvino, Russia) in the Prometheus proton therapy complex, which is a Russian-made high-technology medical complex for the treatment of tumors using the remote radiotherapy technique. This complex is based on a compact proton synchrotron with a beam energy of 30 to 330 MeV [22].

### 4.5. Dosimetry

Experimental verification of the actual peak position and absorbed dose estimation were performed using Gafchromic EBT3 film (Ashland, Wilmington, DE, USA) and PTW dosimetry equipment complex (PTW-Freiburg, Freiburg im Breisgau, Germany): a Unidose webline electrometer with PinPoint 3D Chamber TM31022 and BraggPeak Chamber TM34073 ionization chambers. Measurements were carried out with an allowance for the distance to the sensitive volume and the recombination of charged particles outside the chamber, since the measurements were performed in air without a buildup cap. An Epson^®^ 10000XL (Seiko Epson, Suwa, Japan) flatbed photofilm scanner was used to digitize the EBT3 radiochromic film.

### 4.6. Clinical Evaluation and Assessment of the Course of Radiation Dermatitis in Mice

Clinical evaluation of each animal was performed daily for 70 days after irradiation in order to record clinical manifestations of RD. Two independent, double-blinded researchers graded radiation-induced dermatitis and hyperpigmentation after irradiation according toxicity criteria to the Radiation Therapy Oncology Group (RTOG) [33]:

Grade 0: No change;

Grade 1: Faint erythema, dry desquamation, epilation, decreased sweating;

Grade 2: Tender or bright erythema, moderate edema, patchy moist desquamation;

Grade 3: Moist desquamation in areas other than in skin folds, pitting edema;

Grade 4: Ulceration, hemorrhage, necrosis;

Grade 5: Death.

Animals were photographed from the dorsal side once a week using a camera. The following settings were used: aperture, f 1.8, sensitivity, ISO 320; shutter speed, 1/262 s. The focusing was performed automatically. The shooting scale was 1:2. A tripod with an LED lamp was used for even lighting in a dark room without natural light. Photographs were cataloged, reviewed, and processed using ImageJ software v1.53t (National Institutes of Health, Bethesda, MD, USA).

### 4.7. Histology

Skin samples were taken on 14th day after exposure to proton radiation from both control (unexposed) and experimental animals. For light microscopy, skin samples were fixed in a 4% paraformaldehyde solution, dehydrated with gradient alcohols, and embedded in paraffin according to the standard histological protocols. The sections were stained with hematoxylin and eosin, and the images were captured with an Axiovert 200 microscope, 40× magnification (Carl Zeiss, Oberkochen, Germany). The skin sections were examined for morphology via quantification of the number of follicular units, and assessment of epidermal and dermal thickening, epithelial hyperplasia, follicular epithelial hyperplasia and parakeratotic hyperkeratosis was performed using ImageJ v1.53t (National Institutes of Health, Bethesda, MD, USA). A minimum of 10 fields for each skin slide were analyzed. The thickness of epidermis and dermis was measured using scale bars. The epithelial hyperplasia and parakeratotic hyperkeratosis were scored as “0” (no/minimal changes), “1” (moderate), “2” (marked), or “3” (strong).

### 4.8. Vital Signs Monitoring in Animals

Animals were weighed immediately before irradiation and then on the 7th and 21st days and on the day immediately before euthanasia (day 70).

Blood samples were taken from the animals by a tail tip incision above the clear vein before irradiation and then on the 7th and 21st days after irradiation and on the day immediately before euthanasia (day 70). Blood samples were taken for analysis on a DH36 Vet hematology analyzer (Dymind, Shenzhen, China) to measure the following parameters: white blood cell, lymphocyte, granulocyte, and platelet count. In addition, animal mortality within 30 days after a single exposure to local proton radiation was assessed, and pathomorphological examination of internal organs (spleen, thymus) was performed, including isolation, morphological assessment, and weighing.

### 4.9. Statistical Analysis

The experimental data are presented as the mean ± standard deviation (M ± SD). The statistical significance of differences between the values in experimental groups was determined using the Mann–Whitney U test and Student’s *t*-test. Differences were considered statistically significant at *p* ≤ 0.05. The obtained data were processed using GraphPad Prism 8.0.1 (GraphPad Software, Boston, MA, USA), Microsoft Excel 2016 (Microsoft, Redmond, Washington, DC, USA), and ImageJ v1.53t (National Institutes of Health, Bethesda, MD, USA) software.

## 5. Conclusions

An in vivo model of RD induced using a single local proton beam irradiation with an energy of 88 MeV was developed. Optimal conditions, including proton beam parameters, animal fixation, positioning, and depilation method, were selected during model development. The developed protocol of irradiation in the dose range of 30 to 50 Gy allowed the establishment of an RD model in mice with a short latent period (7–11 days), pronounced stages of formation, characteristic histological skin changes, and no impact on animal well-being (100% survival rate, body weight increase, no changes in the number of formed elements of the blood or relative mass of lymphoid organs). We think that the optimal model for evaluation of pathogenesis and testing of new preventative and therapeutic agents for proton-induced RD is Grade 2 to early Grade 3, obtained at a dose of 30 Gy in nondepilated animals. At this stage, the external manifestations of RD are more pronounced, while the toxicity only affects the epidermis and dermal elements without disrupting the integrity of the basal membrane.

## Figures and Tables

**Figure 1 ijms-24-16373-f001:**
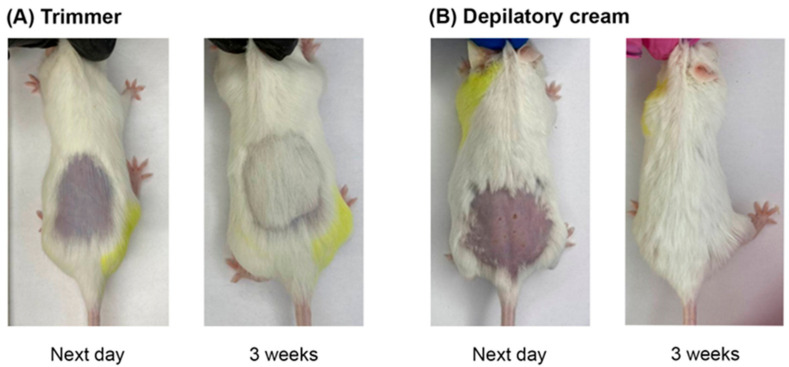
Animal skin the next day and 3 weeks after depilation using (**A**) trimmer; (**B**) depilatory cream.

**Figure 2 ijms-24-16373-f002:**
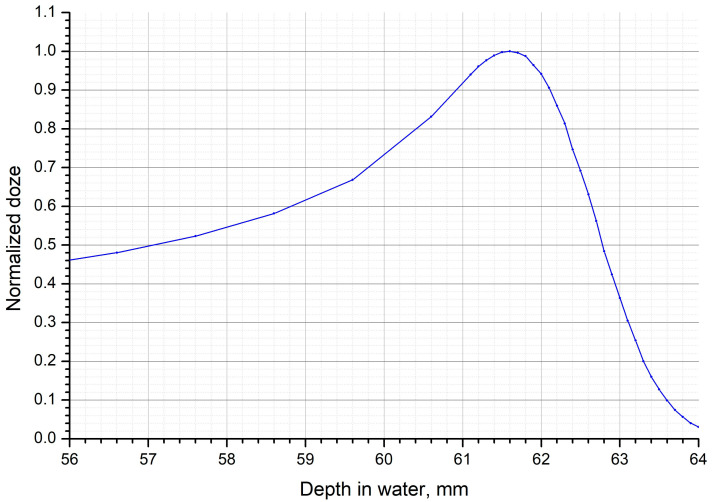
Characteristic dose distribution in water in the Bragg peak region at a proton energy of 88 MeV.

**Figure 3 ijms-24-16373-f003:**
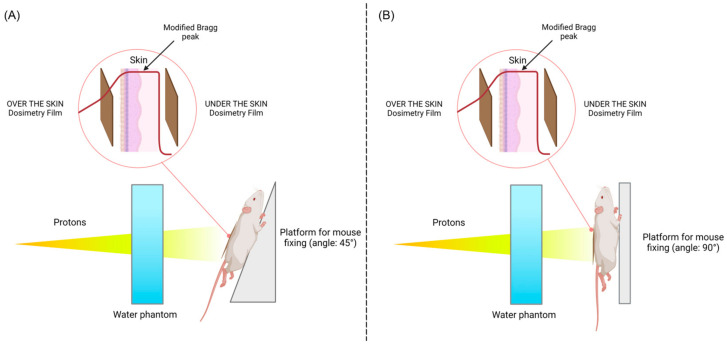
Animal positioning by (**A**) 45°; (**B**) 90°.

**Figure 4 ijms-24-16373-f004:**
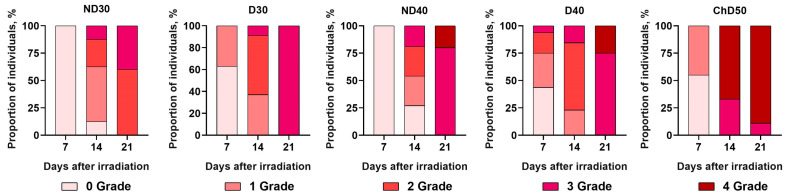
Time-dependent changes in the skin of mice after treatment with 30 Gy (ND30, D30), 40 Gy (ND40, D40), or 50 Gy (ChD50) of proton irradiation on 7th, 14th, and 21th days.

**Figure 5 ijms-24-16373-f005:**
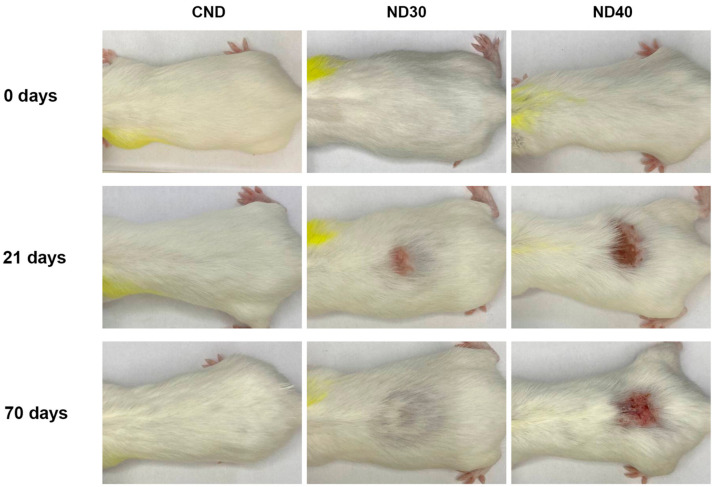
Representative images of spontaneous wound healing of radiation-induced skin damage in nondepilated mice after proton irradiation with a dose of 0 Gy (CND), 30 Gy (ND30) and 40 Gy (ND40).

**Figure 6 ijms-24-16373-f006:**
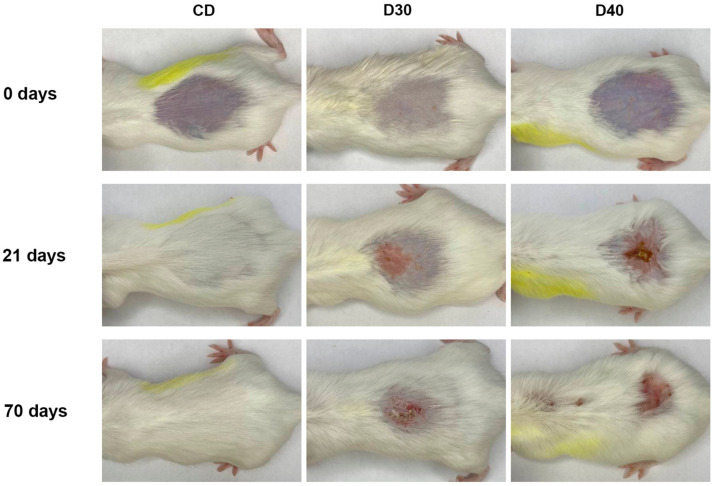
Representative images of spontaneous wound healing of radiation-induced skin damage in physically depilated mice after proton irradiation with a dose of 0 Gy (CD), 30 Gy (D30), and 40 Gy (D40).

**Figure 7 ijms-24-16373-f007:**
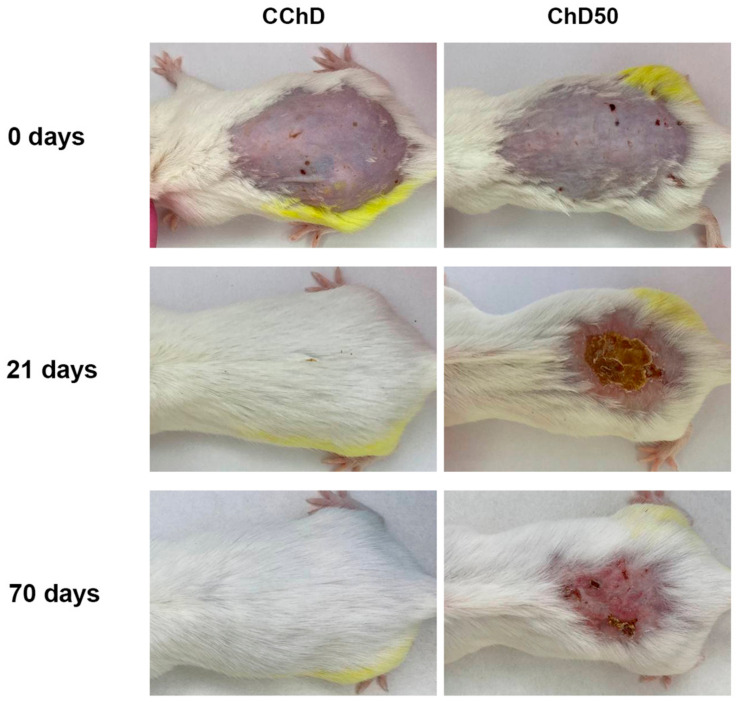
Representative images of spontaneous wound healing of radiation-induced skin damage in chemically depilated mice after proton irradiation with a dose of 0 Gy (CChD) and 50 Gy (ChD50).

**Figure 8 ijms-24-16373-f008:**
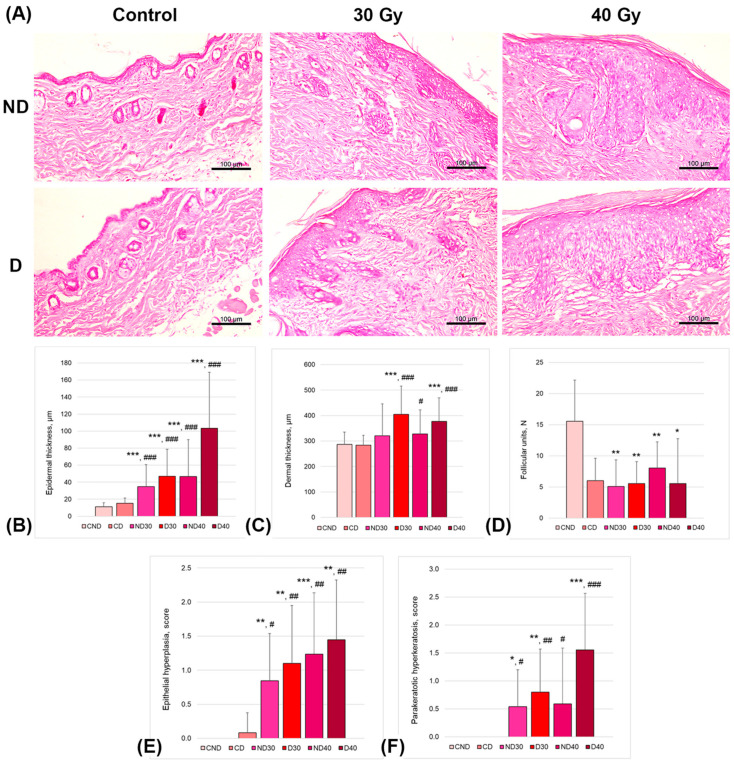
Depilation effects on skin histology in a proton-beam-radiation-induced dermatitis model. (**A**) Representative images of nondepilated (ND) and depilated (D) mice skin samples in control and irradiated groups (30 and 40 Gy), hematoxylin and eosin staining, scale bars 100 µm. The results of (**B**) assessment of epidermal thickness; (**C**) assessment of dermal thickness; (**D**) the number of follicular units; (**E**) scores for epithelial hyperplasia; (**F**) scores for parakeratotic hyperkeratosis. Data are shown as M ± SD and were analyzed using the Mann–Whitney U-test (* *p* < 0.05, ** *p* < 0.005, *** *p* < 0.0005 vs. CND group; # *p* < 0.05, ## *p* < 0.005, ### *p* < 0.005 vs. CD group). CND—control nondepilated animals; CD—control animals depilated using physical method; ND30—non-depilated animals that were irradiated with a dose of 30 Gy; D30—depilated animals that were irradiated with a dose of 30 Gy; ND40—nondepilated animals that were irradiated with a dose of 40 Gy; D40—depilated animals that were irradiated with a dose of 40 Gy.

**Figure 9 ijms-24-16373-f009:**
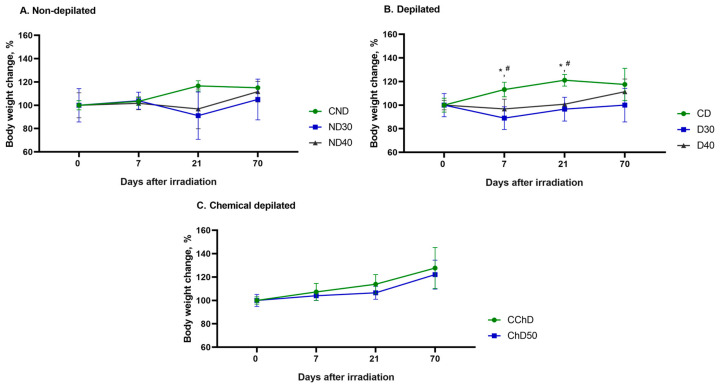
Body weight change in white male mice in RD model induced via single exposure to local proton radiation. Data are shown as M ± SD (%) and were analyzed using Mann–Whitney U-test. (**A**) Relative body weight of mice in % of nondepilated mice (CND—control nondepilated animals, ND30—nondepilated animals that were irradiated with a dose of 30 Gy, ND40—nondepilated animals that were irradiated with a dose of 40 Gy). (**B**) Relative body weight of mice in % of physical method depilated mice (CD—control animals depilated using physical method, D30—depilated animals that were irradiated with a dose of 30 Gy, D40—depilated animals that were irradiated with a dose of 40 Gy) (* *p* < 0.05, D30 vs. CD; ^#^ *p* < 0.05, D40 vs. CD). (**C**) Relative body weight of mice in % of chemical method depilated mice (CChD—control animals depilated using chemical method, ChD50—depilated animals that were irradiated with a dose of 50 Gy).

**Figure 10 ijms-24-16373-f010:**
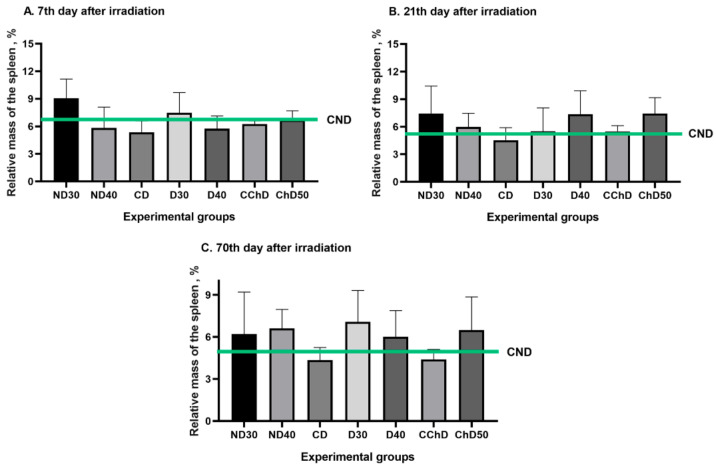
Relative mass of the spleen to the weight change in white male mice in modeling RD induced by single exposure to local proton radiation (CND—control nondepilated animals, ND30—nondepilated animals that were irradiated with a dose of 30 Gy, ND40—nondepilated animals that were irradiated with a dose of 40 Gy, CD—control animals depilated using physical method, D30—animals depilated using physical method that were irradiated with a dose of 30 Gy, D40—animals depilated using physical method that were irradiated with a dose of 40 Gy, CChD—control animals depilated using chemical method, ChD50—animals depilated using chemical method that were irradiated with a dose of 50 Gy). Data are shown as M ± SD (%). (**A**) Relative mass of the spleen in % of mice on 7th day after irradiation. (**B**) Relative mass of the spleen in % of mice on 21th day after irradiation. (**C**) Relative mass of the spleen in % of mice on 70th day after irradiation.

**Figure 11 ijms-24-16373-f011:**
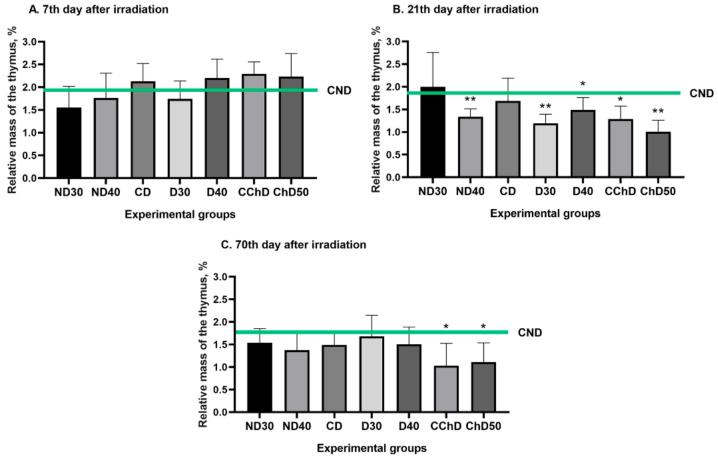
Relative mass of the thymus to the weight change in white male mice in modeling RD induced using single exposure to local proton radiation (CND—control no-depilated animals, ND30—nondepilated animals that were irradiated with a dose of 30 Gy, ND40—nondepilated animals that were irradiated with a dose of 40 Gy, CD—control animals depilated using physical method, D30—animals depilated using physical method that were irradiated with a dose of 30 Gy, D40—animals depilated using physical method that were irradiated with a dose of 40 Gy, CChD—control animals depilated using chemical method, ChD50—animals depilated using chemical method that were irradiated with a dose of 50 Gy). Data are shown as M ± SD (%) and were analyzed using Mann–Whitney U-test (* *p* < 0.05, ** *p* < 0.01 vs. CND). (**A**) Relative mass of the thymus in % of mice on 7th day after irradiation. (**B**) Relative mass of the thymus in % of mice on 21th day after irradiation. (**C**) Relative mass of the thymus in % of mice on 70th day after irradiation.

## Data Availability

Data are contained within the article or Appendix A.

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
