# Peer review of "An Experimental Model of Proton-Beam-Induced Radiation Dermatitis In Vivo"

_ijms, 2023, doi:10.3390/ijms242216373_

Round 1
Reviewer 1 Report
Comments and Suggestions for Authors
This manuscript is presenting a setup and method for evaluation of radiation induced skin toxicity using protons as radiation modality.
The manuscript is indicating with the title, and is also stating in the text, that there is yet no validated radiation dermatitis models for protons. It is unclear what is meant with validated here, but mouse models of skin toxicity after proton radiation already do exists (eg DOI: 10.1080/0284186X.2023.2246641, DOI; 10.1080/0284186X.2023.2254481).
The setup is taking advantage of the proton Bragg peak for irradiating skin, and avoid radiation to any deeper lying tissues or organ structures. That does give some advantages, eg thorax can be irradiated without causing any damage to essential organs. However, the disadvantage is that the LET, which is completely ignored in the paper, cannot be modulated with this setup, and for radiobiological purposes, testing the impact of variable LET values is one of the most important factors.
Q1: The rationale for the selected doses is based on the cumulative dose of fractionated xray therapy (P7 l 278:), but this is not at all comparable, as the biological effect of fractionated radiotherapy is completely different than a single fraction treatment (eg see in the Textbook Basic Clinical Radiobiology, Joiner and van der Kogel). Also in this section, the RBE is used as explanation for the included dose range, but more details need to be added here. The current clinically used RBE for protons is set to 1.1 – is a higher RBE expected in the current study?
Q2: The timing of the evaluation of the skin damage is explained by that “presumably up to 21 days from the moment of proton radiation exposure there is a destruction stage” (p7 l 285) and then a reference (Ref 30, Grebenjuk et al) which cannot be localized? This does not correlate to classic radiobiology, where early skin reaction is a well-defined endpoint, used in a large range of studies for demonstrating changes in radiosensitivity between different modalities and combination of treatments. The biology of skin toxicity is well described (eg in Radiation and Skin, Taylor & Francis, London (1985), pp. 102-152), and is characterized by reddening of the skin in the least severe end of the scale, and by moist desquamation in the most severe end of the scale. The timing is very well defined with the main erythema wave beginning about the tenth day after treatment and increasing to reach a peak around day 14, and the intensity is dose dependent. The dominating process behind the observed moist desquamation is epidermal cell death and the consequent reduction in epidermal cellularity and reduced number of cell layers, leading to reduced epidermal thickness. It also appears that the data in fig 4 would demonstrate this time dependency is presented in a clearer manner (see next comment).
Q3: The treatment groups are quite varying regarding dose and treatment, which makes comparison hard. There is no 50Gy groups which is comparable to 30 or 40Gy, as they have been treated differently. This does not allow for the dose dependent response to be extracted. Fig 4 is very hard to deduct data from. I would recommend the data to be presented differently: One figure with the time dependency of the individual toxicity levels (so four panels, and one curve for each treatment), No of days after irradiation on the x-axis and pct of animals on the y-axis. And a dose dependency figure, where the maximum tox grade is shown as function of dose.
Q3: Fig 8, which time point is this data obtained?
Author Response
Responses to comments of Reviewer #1
We are grateful to the Reviewer #1 for a careful evaluation of our manuscript and valuable comments. Below we give our point-by-point response to Reviewer’s comments (the Reviewer’s remarks are presented in plain text, our answers follow in blue text).
REVIEWER 1
(x) I would not like to sign my review report
( ) I would like to sign my review report
Quality of English Language
(x) I am not qualified to assess the quality of English in this paper
( ) English very difficult to understand/incomprehensible
( ) Extensive editing of English language required
( ) Moderate editing of English language required
( ) Minor editing of English language required
( ) English language fine. No issues detected
|
Yes |
Can be improved |
Must be improved |
Not applicable |
|
|
Does the introduction provide sufficient background and include all relevant references? |
( ) |
( ) |
(x) |
( ) |
|
Are all the cited references relevant to the research? |
( ) |
(x) |
( ) |
( ) |
|
Is the research design appropriate? |
( ) |
( ) |
(x) |
( ) |
|
Are the methods adequately described? |
( ) |
( ) |
(x) |
( ) |
|
Are the results clearly presented? |
( ) |
( ) |
(x) |
( ) |
|
Are the conclusions supported by the results? |
( ) |
( ) |
(x) |
( ) |
Comments and Suggestions for Authors
This manuscript is presenting a setup and method for evaluation of radiation induced skin toxicity using protons as radiation modality.
The manuscript is indicating with the title, and is also stating in the text, that there is yet no validated radiation dermatitis models for protons. It is unclear what is meant with validated here, but mouse models of skin toxicity after proton radiation already do exists (eg DOI: 10.1080/0284186X.2023.2246641, DOI; 10.1080/0284186X.2023.2254481).
We would like to express our gratitude to the reviewer for providing references to new relevant studies on radiation dermatitis induced by proton irradiation, in mice. At the time of submission of this paper, the authors did not have the opportunity to review and cite the data from these studies (DOI: 10.1080/0284186X.2023.2246641, DOI: 10.1080/0284186X.2023.2254481). The findings of these studies are of interest to us and have been included in the text of this manuscript in lines 59-62: «Furthermore, this year saw the publication of unique studies describing dosimetric characteristics and analysis of acute and late effects of normal tissue damage in response to single and fractionated proton irradiation [24,25]». It is important to note that we did not find a reproducible model of acute radiation dermatitis induced by a single localized proton exposure, taking into account parameters such as position of the energy release peak of the monoenergetic proton beam, positioning and fixation of animals, method of depilation or its absence, duration of latent period and stages of radiation dermatitis, model reproducibility (% of animals with expected parameters of RD), assessment of vital signs (weight change, hematological blood test, pathomorphological examination of internal organs) and animal survival, in the literature. We believe that our results contribute to the investigation of the effects of proton radiation on the skin and provide a basis for evaluating the effectiveness of preventive and therapeutic agents in a proton-induced radiation dermatitis model. The timing of RD formation in the proposed model allows for the study of the pathogenesis of radiation-induced skin damage and the assessment of the effectiveness of therapeutic agents in reducing skin damage and accelerating its recovery within a more convenient and compact timeframe compared to fractionated irradiation. This eliminates the need for a prolonged period between therapeutic irradiation and the formation of a desired degree and stage of RD, before the application of the investigational drugs. Additionally, the positioning of the target area in the center of the animal's back minimizes further interventions by the mouse in the application process of the therapeutic agent and wound healing stage.
The setup is taking advantage of the proton Bragg peak for irradiating skin, and avoid radiation to any deeper lying tissues or organ structures. That does give some advantages, eg thorax can be irradiated without causing any damage to essential organs. However, the disadvantage is that the LET, which is completely ignored in the paper, cannot be modulated with this setup, and for radiobiological purposes, testing the impact of variable LET values is one of the most important factors.
As correctly noted by the reviewer, the planning of local skin only irradiation in this study, taking into account the localization and area of exposure, is an advantage. Such a radiation dermatitis model allows for the avoidance of critical organ damage and enables the study of the effects of proton radiation in the treatment of melanoma and basal cell carcinoma, which allows for maximum dose load on the skin with minimal radiation damage to underlying tissues.
We have included information on LET, with a value of approximately 5-10 keV/µm (In lines 151-152: «The value of linear energy transfer (LET) for proton radiation was approximately 5-10 keV/μm.»).
Regarding LET modulation, it was not considered in our experimental setup. We deliberately selected irradiation conditions in the most damaging mode — at the Bragg peak. The irradiation of the skin under a different value of the absorbed proton beam dose is only possible through the entire body of the animal. It will require a modification of the model by pulling up the skin to a distance at which the beam will not pose a risk of overexposure to critically important organs of the experimental animal.
Q1: The rationale for the selected doses is based on the cumulative dose of fractionated x-ray therapy (P7 l 278:), but this is not at all comparable, as the biological effect of fractionated radiotherapy is completely different than a single fraction treatment (eg see in the Textbook Basic Clinical Radiobiology, Joiner and van der Kogel). Also in this section, the RBE is used as explanation for the included dose range, but more details need to be added here. The current clinically used RBE for protons is set to 1.1 – is a higher RBE expected in the current study?
We would like to express our gratitude to the reviewer for their comment. We have made the corresponding changes to the manuscript, which you can read in the revised version. The investigated dosage range of 30 to 50 Gy was selected based on the available literature data on RD modeling using various ionizing radiation sources (In lines 469-480: «…However, in recent decades, the effects of other types of ionizing radiation on the skin have been fairly well described. For example, it was demonstrated that when Balb/c mice were irradiated with a dose of 36 Gy for 3 consecutive days (12 Gy/day) using a source of X-ray radiation, the peak of skin damage was observed 7-10 days after irradiation [19]. In another study [20], using the same mice strain and radiation scheme, the duration of the latent period was 7 days and the peak development of RD occurred on the 15th day. In the study [32], when C57BL/6 mice were irradiated with high-energy electrons at doses of 20 Gy or higher, RD developed 8-12 days after irradiation and progressed to severe grades (3-4) within 3 weeks. In contrast, mice receiving a dose of 15 Gy developed only mild RD. Local irradiation of the flanks of C57BL/6 mice with X-rays at a dose of 45 Gy resulted in severe skin damage, characterized by scab formation and moist desquamation [17]. Subsequently, the authors reduced the irradiation dose to 30 Gy….»).
In addition, the selection of an acute exposure in the dose range of 30 to 50 Gy is justified by the results obtained by our colleagues. It has been shown that there were no significant toxic skin injuries when irradiating the hind limbs of mice with Ehrlich ascites carcinoma with protons at doses of 60 and 80 Gy as a single dose or as two equal doses of 30 and 40 Gy, respectively (https://doi/org/10.33647/2713-0428-17-3E-127-132).
In current study we didn’t investigate the skin toxicity compared to conventional X-irradiation, but it can be assumed that the value of RBE is set to 1.1. When determining the value of the RBE of protons upon irradiation of mice in vivo in the range of low (0.1–1 Gy), therapeutic (2 Gy), and sublethal (4.5–12.5 Gy) doses, our colleagues also determined the RBE value close to 1 upon irradiation before and after the Bragg peak [doi: 10.1134/S1607672920050026, doi: 10.1134/S1607672921040037]. There is a good agreement between the published data in determination of RBE values, which were obtained both in vivo and in vitro [doi: 10.1016/j.mrgentox.2015.08.003].
Q2: The timing of the evaluation of the skin damage is explained by that “presumably up to 21 days from the moment of proton radiation exposure there is a destruction stage” (p7 l 285) and then a reference (Ref 30, Grebenjuk et al) which cannot be localized? This does not correlate to classic radiobiology, where early skin reaction is a well-defined endpoint, used in a large range of studies for demonstrating changes in radiosensitivity between different modalities and combination of treatments. The biology of skin toxicity is well described (eg in Radiation and Skin, Taylor & Francis, London (1985), pp. 102-152), and is characterized by reddening of the skin in the least severe end of the scale, and by moist desquamation in the most severe end of the scale. The timing is very well defined with the main erythema wave beginning about the tenth day after treatment and increasing to reach a peak around day 14, and the intensity is dose dependent. The dominating process behind the observed moist desquamation is epidermal cell death and the consequent reduction in epidermal cellularity and reduced number of cell layers, leading to reduced epidermal thickness. It also appears that the data in fig 4 would demonstrate this time dependency is presented in a clearer manner (see next comment).
We agree with the reviewer that typically reactions to ionizing radiation are observed 10-14 days after exposure. However, we want to emphasize that the peak development of radiation dermatitis does not always occur at 14 days, which is determined by the type of ionizing radiation, absorbed dose, hair removal method, and other factors. We have removed the sentence "presumably up to 21 days from the moment of proton radiation exposure there is a destruction stage" from the manuscript text. However, we want to clarify why the degree of radiation dermatitis was assessed up to 21 days. Firstly, after 21 days, radiation-induced skin damage was observed in all experimental groups, which did not progress over time. Secondly, due to the beginning process of skin repair, a scab forms on the skin, which makes it difficult to assess the degree of radiation dermatitis according to the clinical scale.
Q3: The treatment groups are quite varying regarding dose and treatment, which makes comparison hard. There is no 50Gy groups which is comparable to 30 or 40Gy, as they have been treated differently. This does not allow for the dose dependent response to be extracted.
Regarding the selected dose range and differences in depilation methods, it should be noted that the primary goal of this pilot study was not to obtain a classic dose-response relationship for the influence of proton radiation on toxic skin reactions in mice, as is typically done in radiobiology. It is known from literature that the biological effect of ionizing radiation depends on the absorbed dose; on average, this relationship is direct, with an increase in dose leading to a greater effect. We expected that at a dose of 30 Gy, radiation-induced skin damage would be minimal, and therefore, we attempted to evaluate several irradiation parameters that could contribute to the degree and dynamics of RD formation.
Fig 4 is very hard to deduct data from. I would recommend the data to be presented differently: One figure with the time dependency of the individual toxicity levels (so four panels, and one curve for each treatment), No of days after irradiation on the x-axis and pct of animals on the y-axis. And a dose dependency figure, where the maximum tox grade is shown as function of dose.
Thank you for the suggestion. In the revised manuscript, we have modified the data representation in Figure 4 and included it in the section labeled «2. Results. 2.4 Clinical changes in the skin of mice after local proton radiation».
Figure 4. Time-dependent changes in the skin of mice after treatment with 30 Gy (ND30, D30), 40 Gy (ND40, D40), and 50 Gy (ChD50) of proton irradiation on 7th, 14th and 21th day.
If we understood the reviewer's recommendation correctly regarding the graphical representation of the maximum tox grade as a function of dose, unfortunately, such an assessment of the dependency of RD severity directly on dose was not conducted in this pilot study. We believe that it is necessary to increase the number of doses with a smaller step size (15-50 Gy) in order to obtain a representative dependency. Obtaining such data will allow for further complementation and improvement of the interpretation of the results.
Q3: Fig 8, which time point is this data obtained?
Thank you for the suggestion. We have added information about the timing of skin sampling in lines 708-709: «Skin samples were taken on 14th day after exposure to proton radiation from both control (unexposed) and experimental animals».

Reviewer 2 Report
Comments and Suggestions for Authors
The authors nicely present a mouse model of RD. The paper is globally clear and well written. The topic is new and of interest. However, the section "results and discussion" has in my opinion a rather poor content in terms of discussion compared to the results and should be enriched, possibly separating the two sections. Minor changes in the writing style are also advisable (see next section).
Comments on the Quality of English LanguageEnglish is good. However, I would strongly suggest to simplify some sentences and punctuation (for examples avoiding the use of too many commas in a single period).
Just to give some practical examples:
Ex. "Hadron RT achieves efficient targeting of tumor tissue with low impact on peritumoral healthy tissues" rather than "In addition, an enhancement in the efficiency of dose delivery to the target organ (tumor) and a decrease in the impact on the surrounding healthy tissues 50 due to a reduction in dose per course is achieved by the use of hadron radiotherapy (with protons, neutrons, and ions)"
and also "Acute RD (grade 2-4) was obtained with doses of 30, 40 and 50 Gy, either with or without depilation" instead of "Skin reactions, which can be classified as acute RD from Grade 2 to Grade 4, were obtained 21 with single doses of protons of 30 Gy, 40 Gy, and 50 Gy, beam energy of 88 MeV, with and without 22 the use of various methods of depilation."
and "Cancer treatment is currently mostly based on surgery, chemotherapy and radiation therapy" instead of "To date, radiation therapy, surgery, and chemotherapy, and less often immunotherapy, hormonal, and gene therapy, remain the main methods for cancer treatment" (the other mentioned treatments are not relevant for the aim of the paper).
Author Response
Responses to comments of Reviewer #2
We are grateful to the Reviewer #2 for a careful evaluation of our manuscript and valuable comments. Below we give our point-by-point response to Reviewer’s comments (the Reviewer’s remarks are presented in plain text, our answers follow in blue text).
REVIEWER 2
(x) I would not like to sign my review report
( ) I would like to sign my review report
Quality of English Language
( ) I am not qualified to assess the quality of English in this paper
( ) English very difficult to understand/incomprehensible
( ) Extensive editing of English language required
( ) Moderate editing of English language required
(x) Minor editing of English language required
( ) English language fine. No issues detected
|
Yes |
Can be improved |
Must be improved |
Not applicable |
|
|
Does the introduction provide sufficient background and include all relevant references? |
( ) |
(x) |
( ) |
( ) |
|
Are all the cited references relevant to the research? |
(x) |
( ) |
( ) |
( ) |
|
Is the research design appropriate? |
(x) |
( ) |
( ) |
( ) |
|
Are the methods adequately described? |
(x) |
( ) |
( ) |
( ) |
|
Are the results clearly presented? |
(x) |
( ) |
( ) |
( ) |
|
Are the conclusions supported by the results? |
( ) |
(x) |
( ) |
( ) |
Comments and Suggestions for Authors
The authors nicely present a mouse model of RD. The paper is globally clear and well written. The topic is new and of interest. However, the section "results and discussion" has in my opinion a rather poor content in terms of discussion compared to the results and should be enriched, possibly separating the two sections. Minor changes in the writing style are also advisable (see next section).
Thank you for the suggestion. In the revised version of the manuscript, we divided the "results and discussion" section into two parts, and also expanded the discussion section.
Comments on the Quality of English Language
English is good. However, I would strongly suggest to simplify some sentences and punctuation (for examples avoiding the use of too many commas in a single period).
Just to give some practical examples:
Ex. "Hadron RT achieves efficient targeting of tumor tissue with low impact on peritumoral healthy tissues" rather than "In addition, an enhancement in the efficiency of dose delivery to the target organ (tumor) and a decrease in the impact on the surrounding healthy tissues 50 due to a reduction in dose per course is achieved by the use of hadron radiotherapy (with protons, neutrons, and ions)"
and also "Acute RD (grade 2-4) was obtained with doses of 30, 40 and 50 Gy, either with or without depilation" instead of "Skin reactions, which can be classified as acute RD from Grade 2 to Grade 4, were obtained 21 with single doses of protons of 30 Gy, 40 Gy, and 50 Gy, beam energy of 88 MeV, with and without 22 the use of various methods of depilation."
and "Cancer treatment is currently mostly based on surgery, chemotherapy and radiation therapy" instead of "To date, radiation therapy, surgery, and chemotherapy, and less often immunotherapy, hormonal, and gene therapy, remain the main methods for cancer treatment" (the other mentioned treatments are not relevant for the aim of the paper).
We are grateful to the reviewer for the high evaluation of our work. We checked manuscript and corrected English language.

Round 2
Reviewer 1 Report
Comments and Suggestions for Authors
The authors have mostly done a good and thorough work in implementing the comments into the paper, and the overall quality has been improved. While I for sure agree with the authors that the proposed model can have a use in proton radiobiology, I still strongly disagree with the claim that this is the first in vivo model of proton induced dermatitis (Eg in the title).
There may have been parameters included in the testing of the current model, such as the influence of depilation, which have not been included in previous models, but that does not entitle it as the first in vivo model. I will therefore strongly recommend the title to be changed – This could eg be to “An Experimental Model of Proton Beam-Induced Radiation Dermatitis in vivo”.
Author Response
The title of the manuscript has been changed to “An Experimental Model of Proton Beam-Induced Radiation Dermatitis in vivo”